# BMLM: Bidirectional Large Language Model for Multi-Task Spoken Language Understanding: Better and Faster

## Abstract

Autoregressive large language models (LLMs) have achieved notable success in natural language generation. However, their direct application to natural language understanding (NLU) tasks presents challenges due to reliance on fixed label vocabularies and task-specific output structures. Although instruction-following tuning can adapt LLMs for these tasks, the autoregressive architecture often leads to error propagation and significant time costs from uncontrollable output lengths, particularly in token-level tagging tasks. In this paper, we introduce a bidirectional LLM framework (BMLM) for multi-task spoken language understanding, which eliminates the need for training from scratch and seamlessly integrates with existing LLMs, bridging the gap between extensive pre-trained knowledge and the requirements of understanding tasks. Our evaluation on multiple datasets demonstrates that BMLM significantly outperforms state-of-the-art pre-trained language models and autoregressive LLM baselines. Specifically, on the MixATIS and MixSNIPS datasets, BMLM achieves notable improvements of +3.9% and +4.1% in overall semantic accuracy compared to autoregressive baselines. Additionally, we observe a 123x improvement in inference speed for the MixATIS dataset and a 189x enhancement for the MixSNIPS dataset compared to existing generative LLM baselines. We anticipate that this work will provide a new perspective and foundational support for LLM applications in the NLU domain. [1]

## 1 Introduction

Benefiting from extensive training datasets, Large language models (LLMs) (Jiang et al., 2023; Peng et al., 2023; Touvron et al., 2023) have notably accelerated progress in the field of natural language processing (NLP) (Geogle., 2023) tasks by effectively leveraging in-context learning (Hu et al., 2022b; Kavumba et al., 2023). However, many LLMs applications within the NLP domain predominantly focus on natural language generation (NLG). Though natural language understanding (NLU) applications do exist, they primarily employ end-to-end instruction tuning or prompt-based few-shot frameworks (Pan et al., 2023; Yin et al., 2024b). These methodologies encounter challenges in supervised NLU settings, which demand task-specific output structures with fixed label vocabularies. Prompt-based few-shot approaches are limited by input length constraints, while instruction tuning often suffers from catastrophic forgetting. Moreover, effectively managing multi-tasking in NLU with autoregressive LLMs through end-to-end generation is particularly difficult. This difficulty arises from the inability of these methods to generate cohesive outputs across multiple tasks, primarily due to their inherently sequential nature.

Spoken Language Understanding (SLU) is a subset of NLU, and plays a crucial role in task-oriented dialog systems, with the primary goal of constructing a semantic frame that encapsulates the user's request. This semantic frame is meticulously crafted through intent detection, identifying the user's intentions, and slot filling, extracting pertinent semantic elements. Considering the inherent interrelation between these two sub tasks (Tur & Mori, 2011), premier SLU systems employ joint models to effectively capture this correlation (Goo et al., 2018; Qin et al., 2019). In practical scenarios, it is common for users to convey multiple intents within one utterance (Gangadharaiah &

---

[1] Our code and data is included in the supplementary material.

Narayanaswamy, 2019), which has steered an increasing volume of research to tackle the intricacies of multi-intent SLU. Xu & Sarikaya (2013) and Kim et al. (2017) first established the platform for this investigation. Subsequent works by Qin et al. (2020; 2021b) exploited graph attention networks to model the complex intent-slot interactions, while Huang et al. (2022) introduced chunk-level intent detection framework (CLID) to segment multi-intent utterances at transition points. Furthermore, Yin et al. (2024a) proposed an innovative joint multi-view intent-slot interaction framework (Uni-MIS) to further focus on the role of fine-grained intent in guiding slot filling. Although pre-trained language model (Devlin et al., 2019; Liu et al., 2019b) (PLM)-based frameworks (e.g., Uni-MIS) have demonstrated promising results. However, due to their relatively restricted scale, there exists a compelling case for enlarging the model size and incorporating LLMs that carry a wealth of pre-training knowledge. More recently, Yin et al. (2024b) have developed entity slots explicitly designed to fine-tune LLMs for SLU tasks, although the approach still relies predominantly on autoregressive generation, which may lead to error propagation and increased inference time. In response to these challenges, we introduce a uniquely devised bidirectional large Language model multi-task framework (BMLM) for multi-task SLU applications. Our exhaustive evaluation using 4 widely used multi-task SLU datasets demonstrates that our approach significantly outperforms existing state-of-the-art (SOTA) models, including traditional PLM-based baselines and end-to-end LLM generative baselines. Our model not only reaches superior performance levels but also assures quicker inference times compared to prevailing end-to-end generation LLM methodologies.

To summarize, our contributions can be outlined as follows: (1) We introduce a bidirectional large language model framework for multi-task spoken language understanding. Unlike traditional autoregressive frameworks, BMLM enhances the utilization of whole-context information and learning dynamics in fixed-label vocabulary tasks. (2) Comprehensive tests on 4 widely-used multi-task SLU datasets demonstrated significant improvements of our model over existing SOTA models, including PLM-based and end-to-end generative LLM baseline. (3) BMLM ensures faster inference times compared to current generative LLM methods, increasing the efficiency and practical utility of our LLM-based frameworks.

## 2 RELATED WORK

### 2.1 JOINT INTENT DETECTION AND SLOT FILLING

Joint intent detection and slot filling form the cornerstone of multi-task SLU frameworks, with their notable interdependence catalyzing the development of integrated models to foster synergistic dynamics. Learning paradigms that concurrently address intents and slots have consistently yielded exemplary outcomes. Some methodologies advocating for simultaneous slot filling and intent detection have adopted parameter sharing strategies (Liu & Lane, 2016a; Wang et al., 2018; Zhang & Wang, 2016), while additional approaches explore unidirectional or bidirectional interaction flows (Qin et al., 2021c). Models engaging in unidirectional interaction pathways (Goo et al., 2018; Li et al., 2018; Qin et al., 2019) feature a predominant flow from intent to slot, often utilizing gating mechanisms intricately fashioned for slot filling tasks (Goo et al., 2018; Li et al., 2018). A novel approach by Qin et al. (2019) presents a token-centric intent detection methodology specifically designed to curtail error transmission. On the other hand, bidirectional-flow interaction paradigms (E et al., 2019; Zhang et al., 2019; Liu et al., 2019a; Qin et al., 2021a) consider the reciprocal influences between intent detection and slot filling. A distinguishing study by E et al. (2019) engineered a method that iteratively reinforces both aspects, evidencing mutually beneficial advancements. Ongoing advancements in refining fine-grained intent detection and the interplay of intent-slot interactions have marked a significant progression. Chen et al. (2022) probed into a novel Self-distillation Joint SLU model within a multi-task learning environment, deeming multiple intent detection a weakly supervised problem and tackling it through Multiple Instance Learning (MIL). Meanwhile, Huang et al. (2022) fashioned a chunk-level intent detection technique coupled with an ancillary task to identify intent transition points, thereby optimizing multi-intent recognition accuracy. A noteworthy contribution by Cheng et al. (2023) involved leveraging the transformer architecture to alleviate the intricacies of multi-intent SLU detection tasks. Additionally, the recent efforts by Yin et al. (2024a) presented a joint multi-view intent-slot interaction framework, which emphasizes the guidance of fine-grained intent on slot filling efficacy.

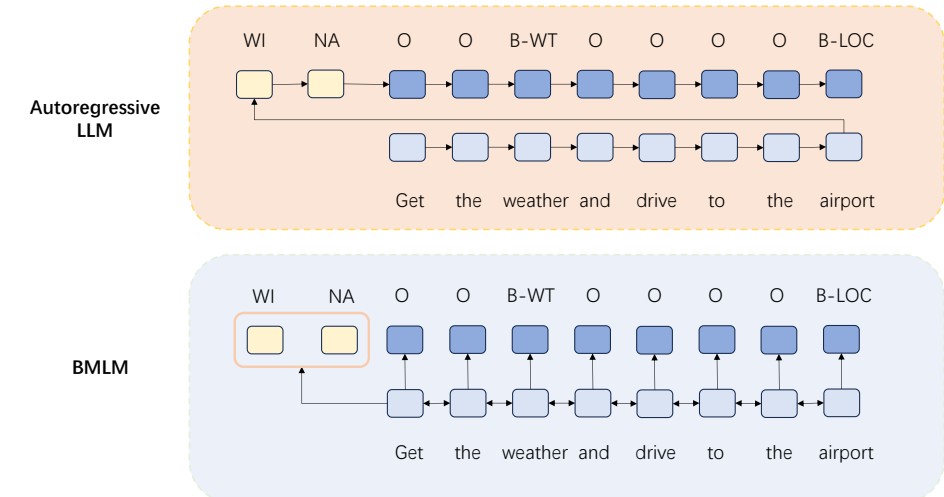

Figure 1: Comparison between BMLM and autoregressive LLMs using a two-intent SLU example. In this context, B-WT represents "B-Weather", B-LOC stands for "B-Location", WI denotes "Weather Inquiry", and NA indicates "Navigation".

## 2.2 LARGE LANGUAGE MODEL FOR NLU TASKS

It is widely observed that the evaluation of LLMs' understanding capabilities frequently utilizes datasets like MMLU (Hendrycks et al., 2021). Although this approach is adequate for assessing the general comprehension abilities of LLMs, it becomes less effective when dealing with label-sensitive NLU tasks that rely on a fixed-label vocabulary. Recent innovations, including the label-supervised Llama framework by Li et al. (2023), have significantly improved the fine-tuning of LLMs for tasks such as named entity recognition (NER). However, these advancements primarily focus on refining LLM capabilities for single, specific tasks. In contrast, multi-task understanding strategies developed by Yin et al. (2024b) leverage LLMs as end-to-end generative models by reshaping the data format used in NLU tasks. This approach presents several benefits. Yet, these models frequently encounter obstacles related to error propagation and prolonged inference times, which are chiefly attributed to their autoregressive configurations.

## 3 APPROACH

This section details the implementation of our proposed methodology. As illustrated in Figure 1, we compare the differences between BMLM and autoregressive LLMs, and we will explain each component of BMLM in the following sections.

### 3.1 PROBLEM DEFINITION

**Intent Detection:** The task of intent detection, given an input sequence $x = (x_1, ..., x_n)$, is framed as a multi-label classification challenge. The goal is to produce a set of intent labels $o_I = (o_1^I, ..., o_m^I)$, where $m$ represents the count of distinct intents within a particular discourse, and $n$ reflects the length of the utterance.

**Slot Filling:** The process of slot filling is akin to a sequence labeling task, which entails mapping the input sequence $x$ to corresponding slot annotations $o_S = (o_1^S, ..., o_n^S)$.

### 3.2 POST-TRAINING CONTEXT-SENSITIVE ATTENTION

In our approach, we introduce a novel modification to the vanilla attention mechanism used in existing LLMs by incorporating context-sensitive attention. This allows the model to retain rich pretrained knowledge and facilitates unrestricted information exchange among all sequence tokens.

Conventional LLMs typically employ a causal mask $\mathcal{M}$ in autoregressive frameworks to prevent future tokens from influencing the generation of present tokens, enforcing a strict left-to-right progression of information flow. In a standard masked attention mechanism, the attention scores $A$ are computed as follows:

$$A = \text{softmax}\left(\frac{QK^T}{\sqrt{d_k}} + \mathcal{M}\right) V \tag{1}$$

where $Q$, $K$, and $V$ represent the query, key, and value matrices, respectively, $d_k$ is the dimension of the key vectors, and $\mathcal{M}$ is the causal mask.

Traditionally, the causal mask $\mathcal{M}$ is defined as:

$$\mathcal{M}_{ij} = \begin{cases} 0 & \text{if } i \geq j \\ -\infty & \text{if } i < j \end{cases} \tag{2}$$

where $i$ represents the position of the token currently attending, and $j$ represents the position of the token being attended to in the sequence.

This limitation can hinder performance in token filling tasks, where understanding the context from both preceding and following tokens is crucial. To address this, we propose an attention mechanism by setting all elements of $\mathcal{M}$ to zero:

$$\mathcal{M}_{ij} = 0 \quad \forall i, j \in \{1, \ldots, n\} \tag{3}$$

where $n$ is the length of the input sequence.

Consequently, the attention computation becomes:

$$A_{\text{context-sensitive}} = \text{softmax}\left(\frac{QK^T}{\sqrt{d_k}}\right) V \tag{4}$$

This adjustment enables the attention mechanism to leverage the entire sequence's context, significantly enhancing token-level representation and addressing limitations imposed by unidirectional flows. By allowing bidirectional context understanding, our approach contributes to a more comprehensive processing of sequences, particularly beneficial for tasks requiring consideration of both past and future contexts.

### 3.3 INTENT DETECTION

Intent detection is treated as a multilabel classification task. After training the model with context-sensitive attention, we employ a **linear classifier** at the final layer to decode the intent tokens, rather than using an autoregressive generation function. This classifier assigns potential intents to each token in the input sequence, which are then selected based on their probability scores to identify the most likely intents, as delineated by the equations:

$$y_I = \text{Intent-Classifier}(\mathbf{H}), \tag{5}$$

$$o_I = \text{Top}_K(y_I), \tag{6}$$

where $\mathbf{H} = \{h_1, h_2, \ldots, h_n\}$ represents the hidden states of the input tokens, $y_I = \{y_1, y_2, \ldots, y_n\}$ denotes the intermediate intent logits generated by the intent classifier, $K$ is the number of intents, and $o_I = \{o_1, o_2, \ldots, o_k\}$ signifies the final predicted intent labels.

### 3.4 SLOT FILLING

Slot filling is approached as a sequence labeling task. Within the context of the BIO (Begin, Inside, Outside) tagging scheme, a **linear classifier** is similarly utilized to tag each input token. This approach accelerates decoding speed and mitigates error propagation during the classification process. This process is succinctly captured by the equation:

$$o_S = \text{Slot-Classifier}(\mathbf{H}), \tag{7}$$

where $\mathbf{H} = \{h_1, h_2, \ldots, h_n\}$ denotes the per-token hidden states derived from the model, and $o_S = \{s_1, s_2, \ldots, s_n\}$ represents the sequence of slot label predictions for each token.

## 3.5 JOINT TRAINING

Unlike autoregressive LLMs, which function as black boxes and do not allow for direct weighting of different tasks, BMLM enables joint optimization for the dual tasks of intent detection and slot filling.

For intent detection, the intent loss $L_{\text{intent}}$ employs the binary cross-entropy formula:

$$L_{\text{intent}} = -\sum_{k=1}^{M} [y_k \log(\sigma(\hat{y}_k)) + (1 - y_k) \log(1 - \sigma(\hat{y}_k))], \tag{8}$$

where $M$ is the total number of intents, $y_k$ is a binary flag indicating the actual presence of the $k$-th intent, $\hat{y}_k$ is the corresponding predictive logit, and $\sigma$ denotes the sigmoid function.

The slot loss $L_{\text{slot}}$ uses the cross-entropy formulation:

$$L_{\text{slot}} = -\sum_{i=1}^{N} \sum_{j=1}^{C} y_{ij} \log(\hat{p}_{ij}), \tag{9}$$

where $N$ represents the total number of tokens in the sequence, $C$ is the number of possible slot classes, $y_{ij}$ indicates the correct classification of token $i$ for slot class $j$, and $\hat{p}_{ij}$ is the model-derived probability that token $i$ belongs to slot class $j$.

The composite loss $L$ synergizes these components, enabling concurrent optimization of both subtasks within a cohesive training algorithm. Additionally, different weight configurations are presented in Appendix A.3:

$$L = L_{\text{intent}} + L_{\text{slot}}. \tag{10}$$

## 4 EXPERIMENTS

### 4.1 DATASETS

Our evaluation extensively utilized two benchmark multi-intent SLU datasets—MixATIS and MixSNIPS (Qin et al., 2021b). MixATIS consists of 13,162 training instances, 756 validation instances, and 828 test instances, primarily focusing on airline-centric queries. In contrast, MixSNIPS spans a broader range of domains, including restaurants and entertainment, comprising 39,776 training instances, 2,198 validation instances, and 2,199 test instances. Both datasets capture realistic complexity in utterances, featuring one to three intents with a 3:5:2 proportional representation. Additionally, experiments were conducted on single-intent datasets, ATIS and SNIPS Coucke et al. (2018); Hemphill et al. (1990), to further validate our model's performance across various settings. The ATIS training set contains 4478 instances, while the test set consists of around 893 instances. In contrast, the SNIPS training set includes about 13,084 instances, and the test set comprises 700 instances.

### 4.2 EXPERIMENTAL SETTINGS

Our experimental setups were carefully designed to maximize training efficiency with the results of the parameter search detailed in the Appendix A.2. We employed Mistral-7B-Instruct-v0.1 (Jiang et al., 2023) as the foundational backbone model for our BMLM model. For fine-tuning, we utilized LoRA (Hu et al., 2022a), setting the LoRA rank at 16 with an alpha scaling parameter of 32, and implemented a dropout rate of 0.05. The optimization regime involved a learning rate of $2 \times 10^{-4}$ and a weight decay of 0.05. Parameter optimization was conducted using the Adam optimizer (Kingma & Ba, 2015). Training steps were adjusted based on dataset size, with 13,162 steps for MixATIS and 39,116 for MixSNIPS.

### 4.3 BASELINES

In the realm of single-intent SLU, notable methodologies include Joint Seq., which offers a multi-task learning architecture integrating domain detection, intent detection, and slot filling within a singular RNN framework (Hakkani-Tür et al., 2016). The Atten.-Based model capitalizes on the attention mechanism to learn correlational dynamics between slots and intents (Liu & Lane,

2016b). Slot-Gated architectures prioritize the mutual dependencies between intent detection and slot filling tasks (Goo et al., 2018). Advanced models such as SF-ID and Stack-Propagation further evolve these principles, with SF-ID introducing explicit connections between slot filling and intent detection, and Stack-Propagation promoting synergetic slot filling guided by intent context (E et al., 2019; Qin et al., 2019). Within the multi-intent SLU landscape, our analysis traversed from the application of the AGIF network in adaptive intent-slot integration to the GL-GIN modules designed for global and local information fusion. We also considered SDJN's multi-task learning strategies and CLID's novel strategy for segmenting complex utterances. Significantly, SSRAN introduced a graph-based approach to deftly navigate the intricate relationships between intents and slots (Qin et al., 2020; 2021b; Chen et al., 2022; Huang et al., 2022; Cheng et al., 2023). Finally, PLM-based methods, such as Uni-MIS (Yin et al., 2024a), along with extensions like Stack-Propagation(Bert), SDJN(Bert) and CLID(Roberta), and generative LLM approach En-Mistral (Yin et al., 2024b), were included to compare the performance of our model against implementations backed by PLM and LLM capabilities.

## 5 EVALUTION

| Model | MixATIS | | | MixSNIPS | | |
|---|---|---|---|---|---|---|
| | Slot(F1) | Intent(Acc) | Overall(Acc) | Slot(F1) | Intent(Acc) | Overall(Acc) |
| AGIF (Qin et al., 2020) | 86.9 | 72.2 | 39.2 | 93.8 | 95.1 | 72.7 |
| GL-GIN (Qin et al., 2021b) | 87.2 | 75.6 | 41.6 | 93.7 | 95.2 | 72.4 |
| SDJN (Chen et al., 2022) | 88.2 | 77.1 | 44.6 | 94.4 | 96.5 | 75.7 |
| CLID (Huang et al., 2022) | 88.2 | 77.5 | 49.0 | 94.3 | 96.6 | 75.0 |
| SSRAN Cheng et al. (2023) | 89.4 | 77.9 | 48.9 | 95.8 | 98.4 | 77.5 |
| SDJN + Bert | 87.5 | 78.0 | 46.3 | 95.4 | 96.7 | 79.3 |
| RoBERTa+Linear | 86.0 | 80.3 | 48.4 | 96.0 | 97.4 | 82.1 |
| CLID + Roberta | 85.9 | 80.5 | 49.4 | 96.0 | 97.0 | 82.2 |
| Uni-MIS Yin et al. (2024a) | 88.3 | 78.5 | 52.5 | 96.4 | 97.2 | 83.4 |
| En-Mistral (Yin et al., 2024b) | 88.7 | 80.6 | 53.4 | 95.6 | 97.6 | 79.8 |
| BMLM (Ours) | 87.4 | 90.5 | **57.3**[*] | 97.2 | 96.1 | **83.9**[*] |

Table 1: SLU performance on MixATIS and MixSNIPS datasets. The most important metric is Overall(Acc). Values with * indicate that the improvement from our model is statistically significant over all baselines ($p < 0.05$ under t-test).

| Model | ATIS | | | SNIPS | | |
|---|---|---|---|---|---|---|
| | Slot(F1) | Intent(Acc) | Overall(Acc) | Slot(F1) | Intent(Acc) | Overall(Acc) |
| Joint Seq (Hakkani-Tür et al., 2016) | 94.3 | 92.6 | 80.7 | 87.3 | 96.9 | 73.2 |
| Atten.-Based (Liu & Lane, 2016b) | 94.2 | 91.1 | 78.9 | 87.8 | 96.7 | 74.1 |
| Sloted-Gated (Goo et al., 2018) | 95.4 | 95.4 | 83.7 | 89.3 | 96.9 | 76.4 |
| SF-ID (E et al., 2019) | 95.8 | 97.1 | 86.9 | 92.2 | 97.3 | 80.4 |
| Stack-Propagation (Qin et al., 2019) | 95.9 | 96.9 | 86.5 | 94.2 | 98.0 | 86.9 |
| Stack-Propagation + BERT | 94.8 | 97.4 | 85.7 | 94.1 | 98.3 | 87.0 |
| En-Mistral | 95.7 | 97.5 | 86.9 | 95.6 | 97.7 | 89.6 |
| BMLM(Ours) | 95.9 | 95.7 | **88.6**[*] | 98.6 | 98.7 | **91.7**[*] |

Table 2: SLU performance on ATIS and SNIPS datasets. Values with * indicate that the improvement from our model is statistically significant over all baselines ($p < 0.05$ under t-test).

### 5.1 MAIN RESULTS

The evaluation metrics included slot F1 score, intent accuracy and semantic accuracy to comprehensively assess the sentence-level semantic frame parsing capabilities. These metrics, adhering to the methodologies delineated by Qin et al. (2021b) and Huang et al. (2022), facilitate a nuanced evaluation of SLU systems. The paramount metric, semantic overall accuracy, quantifies the system's proficiency in simultaneously and correctly predicting both intents and slots within a single sentence. Our results underscore the superior performance of the BMLM, which demonstrates marked improvements in comparison to the autoregressive LLM baseline En-Mistral and other baselines:

(1) As shown in Table 1, on the MixATIS dataset, BMLM achieved a Slot (F1) score of 87.4%, an Intent (Acc) of 90.5%, and an Overall (Acc) of 57.3%. In comparison, the best baseline, En-Mistral,

| Model | MixATIS_Half | | | MixSNIPS_Half | | |
|---|---|---|---|---|---|---|
| | Slot(F1) | Intent(Acc) | Overall(Acc) | Slot(F1) | Intent(Acc) | Overall(Acc) |
| Stack-Propagation | 86.0 | 42.3 | 24.5 | 93.3 | 66.9 | 50.8 |
| AGIF | 86.4 | 67.9 | 37.0 | 93.1 | 93.8 | 68.9 |
| GL-GIN | 86.7 | 75.1 | 40.6 | 93.0 | 94.3 | 69.3 |
| AGIF + Bert | 87.0 | 77.5 | 47.5 | 95.5 | 95.3 | 79.6 |
| GL-GIN + Bert | 84.6 | 81.8 | 48.9 | 95.5 | 94.1 | 80.1 |
| En-Mistral | 84.6 | 78.1 | 46.7 | 95.2 | 96.5 | 77.4 |
| BMLM(Ours) | 89.2 | 79.9 | **51.1** | 96.0 | 97.4 | **81.9** |

Table 3: SLU performance on the MixATIS_Half and MixSNIPS_Half datasets. The Half datasets were constructed based on specific rules, maintaining the label vocabulary from the training set while reducing the data volume by half for analysis.

scored a Slot (F1) of 88.7%, an Intent (Acc) of 80.6%, and an Overall (Acc) of 53.4%. For the MixSNIPS dataset, BMLM attained a Slot (F1) score of 97.2%, an Intent (Acc) of 96.1%, and an Overall (Acc) of 83.9%, surpassing SoTA model Uni-MIS in Overall (Acc), which recorded 83.4%. (2) As shown in Table 2, in the single-intent ATIS dataset, BMLM secured an Overall (Acc) of 88.6%, which is higher than En-Mistral's 86.9%. In the SNIPS dataset, BMLM exhibited robust performance with an Overall (Acc) of 91.7%, also surpassing En-Mistral's 89.6%. (3) As detailed in Table 3, to assess model efficacy under reduced data conditions, we utilized half-sized training datasets—specifically, MixATIS_Half and MixSNIPS_Half. In these constrained environments, BMLM demonstrated resilience, attaining a semantic accuracy of 51.1% on MixATIS_Half and 81.9% on MixSNIPS_Half. Compared to En-Mistral, which achieved Overall (Acc) scores of 46.7% and 77.4% on the respective datasets, BMLM showed a significant improvement in performance.

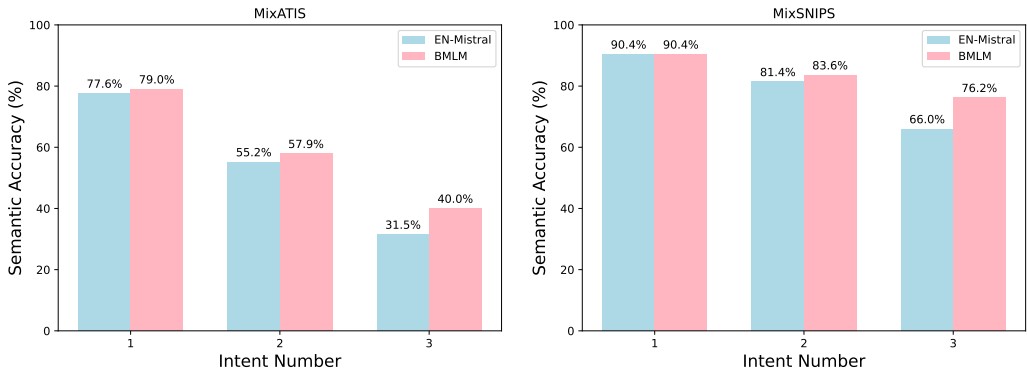

Figure 2: A comparison of the performance of the models on the MixATIS and MixSNIPS datasets, with the data segregated by the number of test instances classified according to the intent.

## 5.2 INFLUENCE OF VARIABLE INTENT NUMBERS

A significant factor impacting model performance in multi-intent SLU tasks is the varying number of intents present within utterances. To gauge this influence, an in-depth evaluation was conducted, segregating instances based on intent number within the MixATIS and MixSNIPS datasets. The details of this categorization are delineated in Figure 2.

Within the MixATIS dataset, EN-Mistral achieved overall accuracies of 77.6%, 55.2%, and 31.5% for utterances with one, two, and three intents, respectively. In contrast, the BMLM model demonstrated superior performance for utterances with two and three intents, recording accuracies of 57.9% and 40.0%. It also slightly outperformed EN-Mistral for single-intent utterances, achieving an accuracy of 79.0%. For the MixSNIPS dataset, the EN-Mistral model reported accuracy scores of 90.4%, 81.4%, and 66.0% for utterances with one, two, and three intents, respectively. In comparison, the BMLM model matched EN-Mistral's performance for single-intent utterances with an accuracy of 90.4%. However, it exhibited modest variations for utterances with two and three intents, achieving accuracies of 83.6% and 76.2%, respectively. This analysis highlights the nuanced performance

characteristics of the BMLM model, particularly its enhanced capabilities in managing complex, multi-intent scenarios within the MixATIS dataset. These comparative assessments underscore the BMLM model's effectiveness in addressing multi-intent SLU tasks, especially in complex scenarios.

## 5.3 IMPACT OF TRAINING DATA PROPORTION

To further investigate the impact of training data proportion, we conducted a comprehensive evaluation, whereby the volume of training data was methodically varied at gradient proportions of 0.2, 0.4, 0.6, 0.8, and 1.0.

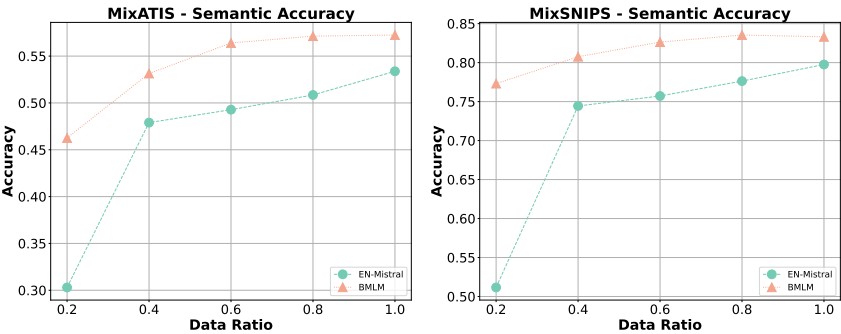

Figure 3: Performance comparison of BMLM and EN-Mistral models on the MixATIS and MixSNIPS datasets at different training data proportions. Semantic accuracy is the focal performance metric in this evaluation.

As shown in Figure 3, in the context of the MixATIS dataset, our assessments distinguished the BMLM model as outperforming the EN-Mistral framework across all proportions of training data. For a randomized data ratio of 20%, BMLM attains a semantic accuracy of 46.3%, significantly outpacing EN-Mistral's performance of 30.3%. This performance advantage persists even as we expand the dataset scope, with BMLM reporting 57.3% semantic accuracy against EN-Mistral's 53.4% upon utilizing the complete training dataset. Regarding the more diverse MixSNIPS dataset, both models exhibit a substantial improvement in semantic accuracy with an increasing volume of training data and BMLM surpasses EN-Mistral across all proportions, initiating at 77.3% versus 51.3% for a 20% data subset and culminating at 83.9% versus 79.8% when leveraging the full dataset.

## 5.4 IMPACT OF DIFFERENT BMLM BACKBONES

As shown in Table 4, the effect of backbone selection is evident across both the MixATIS and MixS-NIPS datasets, with distinct model backbones influencing the datasets differently. For the MixATIS dataset, the PLM backbone RoBERTa achieves a semantic accuracy of 48.4%. In contrast, the Mistral-7B-Base-v0.1 and Mistral-7B-Instruct-v0.1 (default) configurations demonstrate significant improvements, with accuracies of 57.0% and 57.3%, respectively. Notably, the Llama3.1-8B-Instruct configuration outperforms all others, attaining a score of 58.7%. In the context of the MixSNIPS dataset, all tested BMLM backbones exhibit robust performance. The RoBERTa backbone secures a semantic accuracy of 82.1%, while the Vicuna-7B and Mistral-7B-Base-v0.1 structures show slight deficits with accuracies of 80.5% and 84.2%, respectively. The Mistral-7B-Instruct-v0.1 (default) structure follows closely with an accuracy of 83.9%, and the Llama3.1-8B-Instruct achieves an accuracy of 84.4%.

## 5.5 COMPARISON OF INFERENCE EFFICIENCY: BMLM VERSUS EN-MISTRAL

As shown in Figure 4, the inference times for the BMLM and En-Mistral models across the MixATIS and MixSNIPS datasets reveal significant differences in efficiency. Specifically, the En-Mistral model demonstrates inference times of 6653 seconds for MixATIS and 17963 seconds for MixSNIPS, while the BMLM model operates at markedly lower times of 54 seconds and 95 seconds, respectively. This results in an impressive speedup factor of approximately 123.6x for MixATIS and 188.0x

| Model | MixATIS | MixSNIPS |
|-------|---------|----------|
| RoBERTa | 48.4 | 82.1 |
| Vicuna-7B | - | 80.5 |
| Mistral-7B-Base-v0.1 | 57.0 | 84.2 |
| Mistral-7B-Instruct-v0.1 (default) | 57.3 | 83.9 |
| Llama3.1-8B-Instruct | 58.7 | 84.4 |

Table 4: Performance comparison of models with different BMLM backbones on MixATIS and MixSNIPS datasets, measured in terms of semantic accuracy.

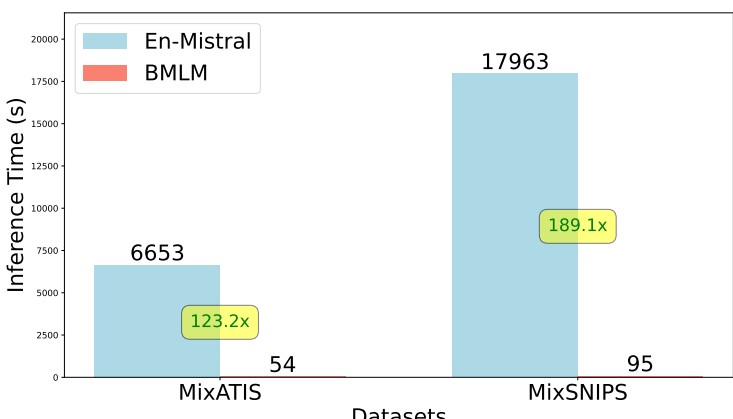

Figure 4: Comparison of inference time with a batch size of 1 on a single RTX 3090 Ti GPU.

for MixSNIPS when comparing BMLM to En-Mistral. Such results highlight BMLM's superior efficiency, making it a compelling choice for real-time applications in spoken language understanding.

### 5.6 CASE STUDIES

To illuminate our framework's efficacy, we delve into a specific instance, as depicted in Figure 6 The scenario "List the Arizona airport and list LA" serves as a prime example. The BMLM hits the mark precisely for both the intents ('atis_airport', 'atis_city') and slots, identifying 'Arizona' as 'B-state_name' and 'LA' as 'B-city_name', perfectly aligning with the ground truth. Conversely, the EN-Mistral model, while precisely predicting the intents, faltered with slots' prediction making an error by categorizing 'Arizona' as part of an 'airport_name'. This implies that the BMLM exhibits more accurate slot tagging in cases where the utterances necessitate attention to a multi-intent scenario. Conversely, the EN-Mistral model evidenced a discrepancy in recognizing the appropriate slots, likely due to its autoregressive nature that may cause it to overlook the necessary clarity required in distinguishing between multi-intent scenarios.

### 6 CONCLUSION

In this study, we have introduced a bidirectional large language model (BMLM) framework aimed at enhancing the performance of multi-task spoken language understanding. This framework represents a significant advancement over traditional pre-trained language models (PLMs) and generative LLM architectures. Through systematic experimentation on four widely-used multi-task SLU datasets, BMLM has achieved state-of-the-art performance, demonstrating a 123x improvement in inference speed on the MixATIS dataset and a 189x enhancement on the MixSNIPS dataset compared to existing generative LLM baselines. Furthermore, BMLM effectively utilizes whole-context information and refines learning processes within fixed-label vocabularies, capitalizing on the extensive knowledge inherent in large language models. These capabilities underscore BMLM's potential for broader

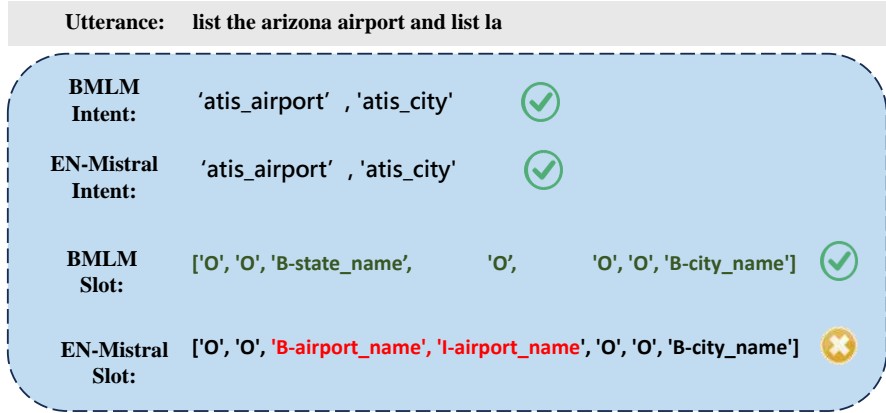

Figure 5: Exemplary comparison of ground truth, BMLM and EN-Mistral's intent and slot predictions for utterance: "List the Arizona airport and list LA". More examples can be found in Appendix A.4.

applications in various natural language understanding tasks, paving the way for future developments in the field.

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

# A APPENDIX

## A.1 LIMITATIONS

The scalability of our model is constrained by computational resources, limiting the BMLM architecture to fewer than 10 billion parameters. This restriction hinders the exploration of larger architectures that may offer improved performance. Additionally, we have not considered the influence of data proportion; specifically, the selection of a representative dataset for training the model. We acknowledge this as an area for future work.

## A.2 PARAMETER SEARCH

| Parameter Setting | Semantic Accuracy (%) | |
|---|---|---|
| | MixATIS | MixSNIPS |
| LoRA Rank = 8 | 56.5 | 81.4 |
| LoRA Rank = 16 | **57.3** | **83.9** |
| LoRA Rank = 32 | 53.6 | 81.6 |
| Learning Rate = 0.01 | 51.2 | 81.9 |
| Learning Rate = 0.02 | **57.3** | **83.9** |
| Learning Rate = 0.03 | 51.6 | 80.9 |

Table 5: Impact of LoRA Rank and Learning Rate on Semantic Accuracy in MixATIS and MixSNIPS datasets.

## A.3 IMPACT OF LOSS WEIGHTING

| Loss Weighting ($\alpha$) | Semantic Accuracy (%) | |
|---|---|---|
| | MixATIS | MixSNIPS |
| Default | **57.3** | 83.9 |
| $\alpha = 0.9$ | 54.0 | 82.2 |
| $\alpha = 0.7$ | 52.4 | 83.4 |
| $\alpha = 0.5$ | 56.0 | **84.0** |
| $\alpha = 0.3$ | 50.7 | 81.9 |
| $\alpha = 0.1$ | 50.1 | 81.7 |

Table 6: Impact of different loss weighting factors ($\alpha$) on Semantic Accuracy in MixATIS and MixSNIPS datasets. The loss is calculated as $L = \alpha L_{intent} + (1 - \alpha)L_{slot}$.

## A.4 MORE EXAMPLES

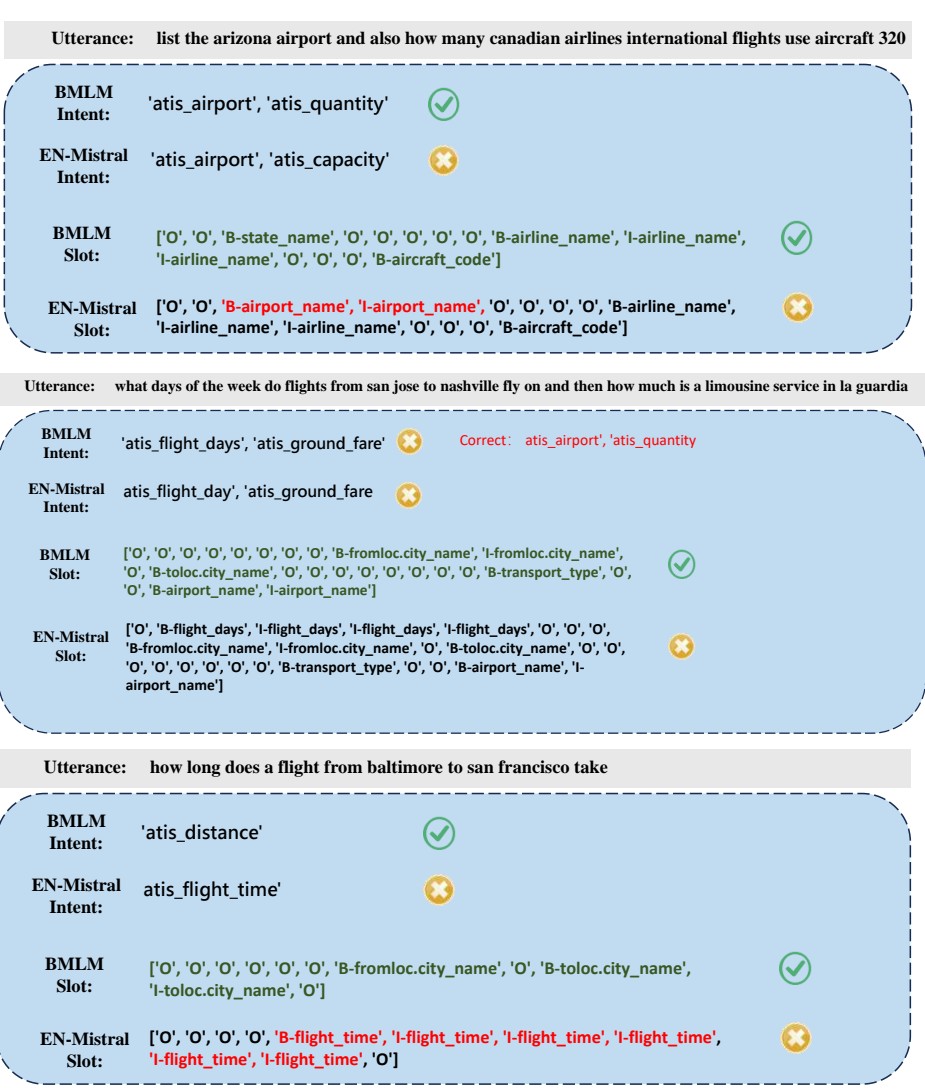

Figure 6: Examples of case studies.