# OpenReview forum: "BMLM: Bidirectional Large Language Model for  Multi-Task Spoken Language Understanding: Better and Faster"
_ICLR.cc/2025/Conference — Submitted to ICLR 2025_

### Official Review · Reviewer_SRtb · 2024-10-17

**Soundness:** 2
**Presentation:** 2
**Contribution:** 2
**Rating:** 3
**Confidence:** 4

**Summary:**

The authors fine-tune LLM (Mistral-7B-instruct) into bidirectional model by removing the causal mask with LoRA. The model is jointly fine-tuned on the tasks of the multi-label IC (sequence classification) and SF (token-level BIO classification) in the common bidirectional way. Experiments are carried out on ATIS, SNIPS, MixATIS, and MixSNIPS with LoRA, and show better results than non-LLMs and LoRA-tuned LLMs.

**Strengths:**

* The paper investigates the under-explored scenario of fine-tuning existing ARLMs into bidirectional models for suitable applications.
* The model reaches good empirical results in a specific task.

**Weaknesses:**

* Although the bidirectional tuning approach can be quite general, applicable to a broad range of sequence tagging tasks, the paper is focused on a rather narrow task. The attractiveness to the broader ICLR community can be limited.
* The proposed method combines multiple components including large-scale models, fine-tuning, and bidirectional tuning. The empirical studies only compare the proposed LoRA-tuned Bi-LLM with a prompted LoRA-tuned LLM and multiple smaller LMs, which can not provide clear support for the contribution and effectiveness of each of the component. In this case, ablation studies can be essential. For example, to determine the aid of the bidirectional tuning, it can be helpful to compare with more approaches to fine-tune LLMs but remain unidirectional, and the specific prompting scheme in En-Mistral complicates this comparison. Possible options include causal generation/language modelling like En-Mistral but without prompting, simply using linear classification heads similar to the proposed method (but remaining unidirectional), repeating the input utterance twice and classifying on the second appearance for a global view, etc. Also, to determine the aid of the model scale, it is better to include the comparison with GPT2-based models under the proposed method, which has the same scale as RoBERTa.
* There are some other works trying to use LLMs in SLU, and it will be better to mention them and compare them with them if possible. Examples include:
https://arxiv.org/abs/2304.04256
https://arxiv.org/abs/2308.14536
https://www.isca-archive.org/interspeech_2023/he23_interspeech.pdf
https://www.isca-archive.org/interspeech_2023/li23s_interspeech.pdf
https://aclanthology.org/2024.lrec-main.1554/
* The speedup in Sec 5.5 is a bit confusing. A long prompt is used in En-Mistral, while it is reasonable to assume that the speed is highly dependent on the length of the input sequence. It would be much better if the average context length used in the experiment were reported for both models to avoid misleading. Also, have you explored using shorter prompts with En-Mistral to potentially improve inference speed while maintaining performance?

**Questions:**

I suggest the authors clarify or address my concerns mentioned above. Also, there are some presentation issues:

* It is better to mark the model sizes in the tables. Particularly, please specify which RoBERTa version (base or large) is used.

* L060~062: "Although pretrained language model...."

* L377~L378: What does "nuanced performance characteristics" here mean? It appears to be quite clear that BMLM outperforms En-Mistral.

---

### Official Review · Reviewer_iM5S · 2024-10-29

**Soundness:** 1
**Presentation:** 2
**Contribution:** 1
**Rating:** 3
**Confidence:** 4

**Summary:**

The paper introduces a bidirectional large language model BMLM to boost performance in multi-task spoken language understanding, which leverages bidirectional context. The model outperforms state-of-the-art baselines on four benchmark datasets, showing substantial gains in semantic accuracy and speed, particularly in multi-intent cases. Additionally, BMLM achieves a significant speedup over generative LLM baselines, enhancing its suitability for real-time SLU applications​.

**Strengths:**

- The paper is readable and well-structured.

- Tackling sequence labeling tasks using a generative framework is indeed challenging. This paper demonstrates the authors' effort in adapting such a framework, which is commendable.

**Weaknesses:**

- The paper lacks recent relevant literature in spoken language understanding (SLU) [1- 4].

- There are significant clarity issues in the methodology. For example, it is unclear how the parameter K in Equation 6 is determined or derived.

- The proposed method’s contributions to the SLU community appear limited. The POST-TRAINING CONTEXT-SENSITIVE ATTENTION primarily involves a simple mask modification to the attention matrix.

- There are major issues in the experimental details that affect the credibility of the results. It is unclear whether the reported table values reflect a single run or an average of multiple runs, which is crucial in understanding the reliability of the results. Additionally, the paper lacks specifics on the t-test procedure: was it applied across multiple model runs, and what confidence intervals were used?

- The experiments lack completeness, particularly concerning ablation studies for the proposed modules.

- Although the authors provide code, it is incomplete and lacks essential documentation, such as clear instructions for running the code and details on the implementation of specific parts of the model.

- The proposed model needs to determine the number of intent within an utterance, but whether a wrong result will impact the prediction of intent and slot labels remains unclear, some more in-depth experiments need to be conducted.

- Existing datasets are relatively simple, it’s easy for LLMs to determine the number of intents within an utterance since there exists an explicit indicator ‘and’ within it. In real application scenarios, there may not be clear boundaries among different intents in user utterances.

- The term "spoken" in "Spoken Language Understanding" could be misleading if the model is not specifically focused on SLU from speech inputs. A more accurate term, such as "natural language understanding" (NLU), may be preferable unless the model genuinely involves handling spoken language input.

- The paper should consider the relevance of standalone NLU modules in the era of large-scale, general-purpose language models. Agent-based frameworks are increasingly capable of handling end-to-end dialogue tasks, reducing the practical utility of isolated NLU components.

[1] Xing B, Tsang I W. Co-guiding for multi-intent spoken language understanding[J]. IEEE Transactions on Pattern Analysis and Machine Intelligence, 2023.

[2] Qin L, Wei F, Chen Q, et al. CroPrompt: Cross-task Interactive Prompting for Zero-shot Spoken Language Understanding[J]. arXiv preprint arXiv:2406.10505, 2024.

[3] Zhuang X, Cheng X, Zou Y. Towards explainable joint models via information theory for multiple intent detection and slot filling[C]//Proceedings of the AAAI Conference on Artificial Intelligence. 2024, 38(17): 19786-19794.

[4] Pham T, Tran C, Nguyen D Q. MISCA: A Joint Model for Multiple Intent Detection and Slot Filling with Intent-Slot Co-Attention[J]. arXiv preprint arXiv:2312.05741, 2023.

**Questions:**

N/A

---

### Official Review · Reviewer_PgxM · 2024-11-09

**Soundness:** 2
**Presentation:** 3
**Contribution:** 1
**Rating:** 3
**Confidence:** 5

**Summary:**

This paper proposes a framework for solving the intent-slot detection tasks in spoken language understanding dialog systems using a bidirectional large language model and joint intent and slot classifiers on top of the LLM. Bidirectionality is introduced during the task specific finetuning phase by removing the causal mask which is typically used in standard auto-regressive LLMs. The authors show that their approach has notable improvements on the MixATIS and MixSNIPS datasets, while also being considerably faster than auto-regressive LLM baselines since all the target intent and slots can be decoded in a single step from the classifiers instead of being generated autoregressively.

**Strengths:**

1. The motivation and proposed approach are explained clearly.
2. Evaluation clearly shows that the proposed approach outperforms baselines, while also being faster.

**Weaknesses:**

1. The fundamental weakness of this paper is novelty. Bidirectional transformers or RNNs, combined with intent and slot tagging classifiers have been used for years. Some references - BERT-based (https://arxiv.org/pdf/1902.10909), RNN-based (https://www.isca-archive.org/interspeech_2016/liu16c_interspeech.pdf). It appears the main contributors to the improved scores shown in this paper are the larger LLMs that are currently available. Discounting those, there isn't any new contribution in the paper. Please clarify if you believe there was a misunderstanding.
2. A major claim in the paper was that the removal of the auto-regressive mask is useful to encode contextual information better for the classifiers. While this makes intuitive sense, this claim definitely warrants an ablation to understand if this is indeed the case, especially given the backbone LLM might be biased to this kind of training. What happens if you keep the rest of your framework fixed, and simply train with a standard auto-regressive mask?

**Questions:**

Both my questions are phrased in the weaknesses section.

---

### Official Review · Reviewer_XfRi · 2024-11-10

**Soundness:** 2
**Presentation:** 2
**Contribution:** 2
**Rating:** 3
**Confidence:** 4

**Summary:**

This paper presents BMLM, a bidirectional large language model framework tailored for multi-task spoken language understanding (SLU). Traditional autoregressive models, though successful in language generation tasks, encounter limitations when applied to SLU, including issues with fixed label vocabularies, error propagation, and inefficiency in handling token-level tasks. To address these, BMLM leverages a bidirectional context-sensitive attention mechanism, enabling efficient multi-task SLU without retraining from scratch and aligning seamlessly with existing large language models.

**Strengths:**

- This paper demonstrates that large language models (LLMs) can effectively handle semantic understanding tasks with bidirectional self-attention through simple post-training.
- Experimental results on various benchmarks and across different data ratios highlight the effectiveness of BMLM.

**Weaknesses:**

While this paper provides valuable insights into LLM capabilities, there are several areas for improvement:

- The paper primarily focuses on spoken language understanding (SLU) tasks, which limits the broader impact of its findings. Validating the model across various semantic understanding tasks could significantly enhance its impact. For example, if the paper were framed around investigating LLMs as “effective bidirectional semantic parsers,” it would provide a more comprehensive and insightful contribution to the field.

- The analysis remains largely task-specific, with a focus on traditional SLU tasks, rather than offering a deeper exploration of findings based on the BMLM itself. It would be beneficial to include analyses such as attention heatmaps post-fine-tuning, the effects of different training setups or datasets on BMLM, and whether results generalize across tasks. This would enrich the study and extend its relevance beyond SLU.

**Questions:**

Given the above weaknesses, could the authors provide additional findings or analyses based on BMLM to help readers better understand the impact of bidirectional post-training? For instance, exploring different tasks or conducting analyses focused on BMLM could help reveal its unique characteristics and broader applicability.

---

### Meta-Review · Area_Chair_ho3Z · 2024-12-12

**Metareview:**

The paper propose a new approach for SLU tasks using bidirectional large language models. However, it lacks novelty (BERT-based models have been widely used for SLU) and has a limited scope (looking at what other tasks can benefit from bidirectional attention). Addressing these weaknesses through more comprehensive analyses, clearer experimental details, and broader task validation could significantly improve the paper's contribution to the field.

**Additional Comments On Reviewer Discussion:**

None

---

### Decision · Program_Chairs · 2025-01-22

Reject